# SKETCHMIND: A Multi-Agent Cognitive Framework for Assessing Student-Drawn Scientific Sketches

**Ehsan Latif**
AI4STEM Education Center
University of Georgia
Athens, GA 30605

**Zirak Khan**
School of Computing
University of Georgia
Athens, GA 30605

**Xiaoming Zhai** *
AI4STEM Education Center
University of Georgia
Athens, GA 30605

## Abstract

Scientific sketches (e.g., models) offer a powerful lens into students' conceptual understanding, yet AI-powered automated assessment of such free-form, visually diverse artifacts remains a critical challenge. Existing solutions often treat sketch evaluation as either an image classification task or monolithic vision-language models, which lack interpretability, pedagogical alignment, and adaptability across cognitive levels. To address these limitations, we present SKETCHMIND, a cognitively grounded, multi-agent framework for evaluating and improving student-drawn scientific sketches. SKETCHMIND introduces *Sketch Reasoning Graphs (SRGs)*, semantic graph representations that embed domain concepts and Bloom's taxonomy-based cognitive labels. The system comprises modular agents responsible for rubric parsing, sketch perception, cognitive alignment, and iterative feedback with sketch modification, enabling personalized and transparent evaluation. We evaluate SKETCHMIND on a curated dataset of 3,575 student-generated sketches across six science assessment items with different highest order of Bloom's level that require students to draw models to explain phenomena. Compared to baseline GPT-4o performance without SRG (average accuracy: 55.6%), and with bSRG integration achieves 77.1% average accuracy (+21.4% average absolute gain). We also demonstrate that multi-agent orchestration with SRG enhances SKETCHMIND performance, for example, a SketchMind with GPT-4.1 gains an average 8.9% increase in sketch prediction accuracy, outperforming single-agent pipelines across all items. Human evaluators rated the feedback and co-created sketches generated by SKETCHMIND with GPT-4.1, which achieved an average of 4.1 out of 5, significantly higher than those of baseline models (e.g., 2.3 for GPT-4o). Experts noted the system's potential to meaningfully support conceptual growth through guided revision. Our code and (pending approval) dataset will be released to support reproducibility and future research in AI-driven education.

## 1 Introduction

Sketching such as drawn models is a fundamental tool in science education, allowing students to externalize their thinking, represent causal mechanisms, and engage in higher-order reasoning [30]. However, assessing the quality and cognitive depth of student-generated sketches remains a long-standing challenge due to their open-ended nature and semantic variability. Automated systems often struggle with interpreting free-form, domain-rich visual input, which makes effective feedback and evaluation particularly difficult. Recent advances in multimodal large language models (MLLMs), such as GPT-4V, have enabled breakthroughs in vision-language reasoning, leading to promising applications in educational assessment [16]. Specifically, systems like NeRiF have shown that

---

*Corresponding author email: `xiaomig.zhai@uga.edu`

39th Conference on Neural Information Processing Systems (NeurIPS 2025).

MLLMs can approximate expert grading of student-drawn models by extracting latent structure from images and comparing them against rubrics [15]. However, such monolithic models still face limitations regarding their reasoning processes, feedback personalization, and inconsistent across conceptually diverse tasks [28, 12].

To address these limitations, we propose SKETCHMIND, a cognitively grounded, multi-agent framework for evaluating and enhancing scientific sketches. SKETCHMIND models sketches as SRGs, which embed both structural semantics and Bloom's Taxonomy-based cognitive annotations [13, 8, 6]. Each SRG encodes domain concepts, their relationships, and Bloom's levels, enabling meaningful alignment with rubrics and scaffolding-targeted feedback. Inspired by the pedagogical principles behind systems like Betty's Brain [3, 18, 4], SKETCHMIND framework decomposes the assessment task across four specialized agents. These agents perform (1) rubric parsing and reference SRG generation, (2) sketch perception and SRG inference, (3) cognitive alignment and scoring, and (4) iterative feedback and sketch modification. This modular design is grounded in cognitive science and agentic learning principles, enabling transparent reasoning and pedagogically informed intervention [14, 9].

Through extensive evaluation on a curated NGSS-aligned dataset of student-drawn science sketches [30], we show that SKETCHMIND not only improves the baseline monolithic MLLM approaches but also increases the capabilities of reasoning models in both quantitative metrics (accuracy and alignment) and qualitative human feedback (clarity, relevance, pedagogical value). Human experts highlighted that SKETCHMIND with models like GPT-4.1 can iteratively improve students' conceptual understanding and sketch quality via visual hints and structured revision cycles. Here are key contributions of this paper:

- We introduce SKETCHMIND, a multi-agent framework that integrates cognitive theories with AI to assess student-generated sketches effectively using our proposed SRGs.

- We integrate Bloom's Taxonomy as cognition theory standard to construct and analyze SRGs for structured evaluation of visual student work and provides pedagogically sound feedback along with real-time sketch modification.

- With empirical studies, we found that SKETCHMIND with State-Of-The-Art MLLM such as GPT-4.1, is able to achieve an average 90.2% sketch prediction accuracy and generate pedagogically sound feedback with sketch modifications highly rated by human experts (4.1 out of 5), highlighting its potential for advanced AI-supported learning of scientific concepts.

To promote transparency and facilitate further research, we have open-sourced our codebase at our repository[2] and plan to make the dataset publicly available upon receiving the necessary approvals. This work represents a step forward in AI for Education, demonstrating how cognitively-aware, agentic systems can advance the quality, transparency, and effectiveness of automated reasoning over student-generated visual content.

## 2 Related Work

**Sketch Understanding and Visual Reasoning.** Recent years have seen significant progress in sketch understanding, particularly within the computer vision community. Approaches such as SketchFusion [2], Sketch2Saliency [2], and SketchXAI [21] have explored the utility of human-drawn sketches for learning visual concepts and providing interpretable representations. These works primarily focus on object detection, image retrieval [5], or 3D modeling [19], rather than assessing the conceptual depth embedded in scientific sketches. Educationally focused sketch models such as SEVA [20] and DrawEduMath [1] analyze human abstraction or math reasoning but lack cognitive scaffolding like Bloom's taxonomy. SKETCHMIND departs from these efforts by representing student-drawn sketches as cognitively annotated semantic graphs and grounding visual elements in educational rubrics, allowing for pedagogical interpretation and targeted feedback.

**Multimodal and Agentic Reasoning in Education.** MLLMs such as GPT-4V have opened new opportunities in visual question answering and diagrammatic reasoning [16]. While tools like NeRiF [15] demonstrate GPT-4V's ability to grade drawn models, these systems often operate as monolithic

---

[2]https://github.com/ehsanlatif/SketchMind

black boxes, limiting transparency and pedagogical adaptability. Similarly, recent work in multimodal chain-of-thought reasoning [30, 27, 29] and multi-agent systems [7, 9, 25, 14] show promise for decomposing complex tasks. However, most frameworks either lack cognitive modeling or fail to integrate sketch modifications explicitly. In contrast, SKETCHMIND brings together modular reasoning agents with fine-grained cognitive alignment, enabling both transparent evaluation and actionable feedback.

**Educational AI for Scientific Sketch Assessment.** Several studies have explored automated grading and classification of hand-drawn sketches in educational settings. Rakhmanov [23] proposed a quality-based classification framework for freehand sketches, while Rahaman et al. [22] applied CNN-based models for accuracy prediction. Lee et al. [17] developed a rubric-driven grading system focused on particulate matter diagrams. These systems, however, primarily emphasize surface-level visual features and often lack semantic or cognitive interpretation. More recent work has tried to integrate learning objectives, such as [28, 12], yet they remain constrained to textual responses or fixed rubrics. SKETCHMIND bridges this gap by introducing SRGs embedded with Bloom-level annotations and enabling multi-agent-driven sketch modification.

## 3 Proposed Approach

We present SKETCHMIND, a cognitively grounded, multi-agent framework for evaluating and improving scientific sketches through iterative, feedback-driven modification. SKETCHMIND is anchored in *Bloom's Taxonomy* [13], a hierarchical model of cognitive processes ranging from recall to creative synthesis. By modeling sketches as semantic structures called SRGs, SKETCHMIND align student-generated content with domain rubrics and provide interpretable, formative feedback across cognitive levels.

### 3.1 Cognitive Framework: Bloom's Taxonomy in Sketch Understanding

Bloom's Taxonomy structures learning objectives into six ascending levels of cognitive complexity [6]:

$$\mathcal{B} = \{\text{Remember}, \text{Understand}, \text{Apply}, \text{Analyze}, \text{Evaluate}, \text{Create}\}. \tag{1}$$

These range from basic recall of knowledge (REMEMBER) to the synthesis of novel ideas (CREATE). This cognitive hierarchy has long served as a foundation in science education for designing assessments and scaffolding learning [13]. Prior work further shows that aligning instructional technologies with Bloom's levels supports measurable gains in higher-order thinking [8]. In SKETCHMIND, scientific sketches are not merely visual representations but are conceptualized as cognitive artifacts that externalize learners' mental models. To operationalize this, we annotate each node (concept) and edge (relation) in the SRG with a Bloom level, a process we call *Bloom-Level Annotation*. This provides a fine-grained measure of the depth of conceptual engagement demonstrated by the student.

To implement this systematically, rubric statements are parsed to extract key verbs and criteria, which are then mapped to Bloom levels using a curated lexicon [24]. The resulting numeric levels (1 for REMEMBER through 6 for CREATE) are encoded as attributes on SRG nodes and edges. These attributes directly support semantic similarity scoring, nuanced evaluation, and feedback generation.

**Pedagogical Integration.** Each gold-standard SRG is labeled at the level of its highest Bloom demand. Student sketches are then evaluated by comparing the Bloom-level annotations of their SRGs against this reference. This supports both diagnostics (identifying the highest level achieved) and adaptive feedback. For example, if a student's sketch demonstrates UNDERSTAND, the system can generate targeted textual feedback or visual hints nudging them toward APPLY. Such progression-oriented scaffolding is consistent with educational research showing that adaptive support aligned with Bloom's hierarchy fosters deeper learning in STEM domains [13, 8].

A schematic illustration (see Fig. 1) depicts this process: a rubric statement (e.g., "Develop a model to explain the transfer of thermal energy") maps to an SRG nodes of (FASTER_MOTION) and (WATER_PARTICLE_HOT), which are then assigned a Bloom level (UNDERSTAND). This visual link between task, rubric, SRG structure, and Bloom's hierarchy enhances transparency for both learners and instructors.

## 3.2 Sketch Reasoning Graphs (SRGs)

We define an SRG as a cognitively annotated semantic graph extracted from a sketch:

$$f_{\text{SRG}}(x) = G = (V, E, \ell, \lambda), \tag{2}$$

where $x \in \mathcal{I}$ is a sketch image, $V \subseteq C$ are concepts from ontology $\mathcal{O} = (Concept, Relation)$, $E \subseteq V \times V$ are directed relations such as causality, and $\ell : V \to \mathcal{B}$ maps each node to a Bloom level. The annotation function $\lambda : V \cup E \to \mathcal{E}$ captures visual and textual evidence ($\mathcal{E}$) supporting the cognitive label.

We treat both the representative gold standard reference sketch for given task $r$ and the student drawn sketch $x$ as inputs to this mapping, producing a reference graph $G_o$ and a student graph $G_s$ respectively. Importantly, the Bloom level annotations are central to how these graphs are constructed and interpreted throughout the system.

## 3.3 Agent Roles

SKETCHMIND comprises four agents, each of which contributes to the construction, interpretation, or refinement of SRGs for a Bloom-aligned assessment prediction:

**Agent 1: Rubric Parser.** Agent 1 in SKETCHMIND performs a static analysis of the representative gold standard reference sketch and rubric for given task $r$ to construct $G_o$, the gold-standard SRG. This process includes explicit mapping of each rubric concept to a Bloom level using expert annotations. For example, components that recall facts are labeled REMEMBER, while those requiring functional understanding or multi-step reasoning are labeled APPLY or ANALYZE. This Bloom-informed rubric becomes the benchmark for evaluating conceptual depth. Figure 1 delineates the SRG creation from the given question, textual rubric and gold standard reference sketch. Agent 1 not only generates gold-standard $G_o$ but also provides reverse mappings $\phi$ to create visuals from cognitive concepts for the target question. These reverse mappings will be used by subsequent agents to provide visual support and sketch modification for improved learning.

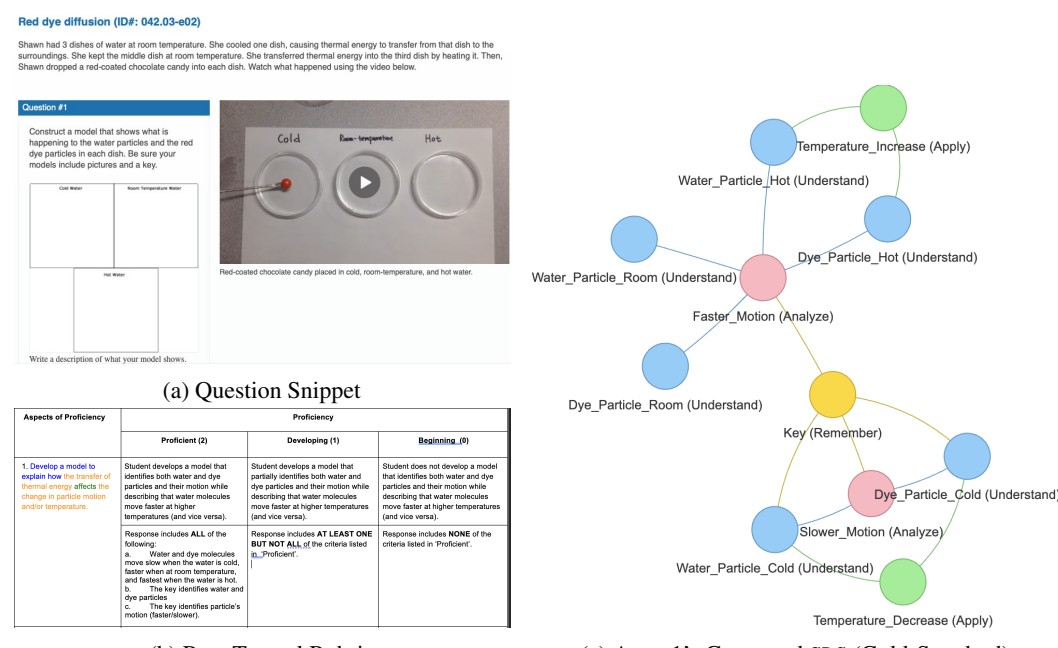

(a) Question Snippet

(b) Raw Textual Rubric

(c) Agent1's Generated SRG (Gold-Standard)

Figure 1: **Overview of Reference SRG Generation.** Given the textual description of question, an expert-designed textual rubric for student sketch performance evaluation, and golden standard reference sketches; Agent 1 processes the information and extracts SRG components and builds Level 4 Bloom's taxonomy ordered SRG (Bloom level ) to set the Gold standard for further evaluation and sketch modification.

**Agent 2: Perception.** Agent 2 in SKETCHMIND applies a MLLM to infer the student SRG $G_s$ from the sketch image $x$. Beyond identifying visual elements, it infers semantic roles and Bloom levels using MLLM directly. For instance, correctly labeling a diagram element might reflect UNDERSTAND, whereas indicating a dynamic interaction (e.g., force, flow) might reflect APPLY or higher. Thus, Agent 2 directly constructs the cognitive structure of the student's mental model. Figure 2 delineates the sample student drawn sketch and Agent 2's perceived SRG, which is then further used for cognitive alignment by Agent 3.

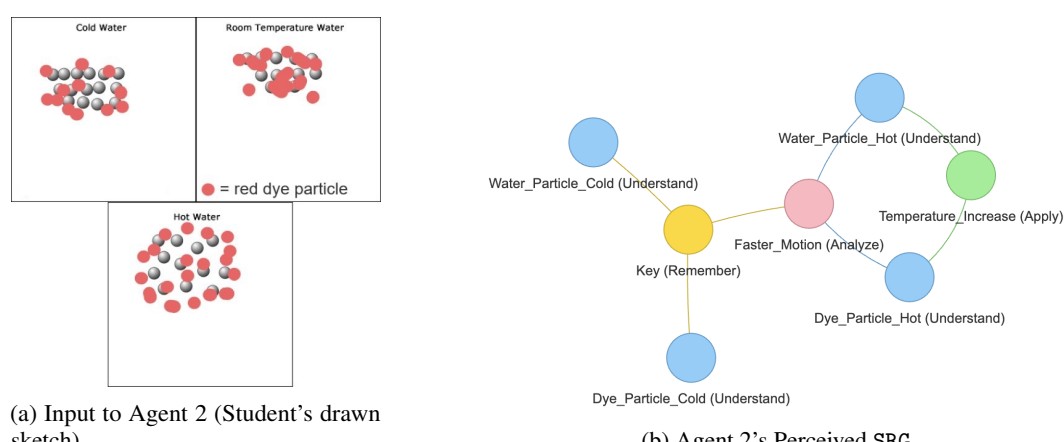

(a) Input to Agent 2 (Student's drawn sketch)

(b) Agent 2's Perceived SRG

Figure 2: Sample sketch drawn by student and Agent 2 to extract perceived SRG. (a) student's drawn sketch, (b) Agent 2's perceived SRG based on the given sketch.

**Agent 3: Cognitive Alignment Evaluator.** Agent 3 in SKETCHMIND compares $G_s$ to $G_o$, computing structural and semantic similarity while analyzing Bloom-level mismatches. To get the similarity score, it first computes the ontology-based node alignment in such a way that for each pair $(v_s, v_o) \in V_s \times V_o$, it calculates weight $w(v_s, v_o)$ as:

$$w(v_s, v_o) = \alpha \cdot \text{sim}_{\mathcal{O}}(v_s, v_o) + (1 - \alpha) \cdot \mathbb{I}[\ell(v_s) = \ell(v_o)], \tag{3}$$

where $\text{sim}_{\mathcal{O}}$ is semantic similarity from ontology $\mathcal{O}$. The summation of these weights is then normalized by the number of total nodes in both graphs to get the overall semantic similarity:

$$f_{OA}(V_s, V_o) = \frac{1}{|V_s + V_o|} \sum_{(v_s, v_o) \in \text{align}} w(v_s, v_o). \tag{4}$$

The similarity score $S \in [0, 1]$ is defined as:

$$S(G_s, G_o) = 1 - \left( \gamma_1 \cdot \frac{f_{GED}(G_s, G_o)}{Z} + \gamma_2 \cdot (1 - f_{OA}(V_s, V_o)) \right), \tag{5}$$

where $f_{GED}$ is function to calculate graph-edit distance (node/edge insertions, deletions, substitutions) noramlized by $Z$ (total number of edges and nodes in both graphs), $f_{OA}$ measures semantic alignment via ontology-based node similarity, and weights $\gamma_1, \gamma_2$ are calibrated on a training set. Notably, both the edit and alignment components consider Bloom-level mismatches as part of the error cost, penalizing regressions in cognitive complexity. It then computes the dominant Bloom level expressed in the sketch as:

$$\hat{y} = \text{mode}\{\ell(v) \mid v \in V_s \cap V_o \iff S(G_s, G_o) > \tau\}, \tag{6}$$

where $\tau$ is the minimum similarity threshold pre-defined to extract overlapping features. This highlights any regression in complexity (e.g., if a concept expected at ANALYZE is represented at only REMEMBER). These mismatches guide the diagnosis of underdeveloped concepts, forming the basis for targeted, Bloom-aligned feedback.

| Similarity Score from Agent 3 and Textual Feedback from Agent 4 |
|---|

**Similarity_score:** 0.592
**Feedback:**
**Your Proficiency Level:** Developing
**What You Did Well:**
The student demonstrated sound structure despite missing some components.
**What Needs Attention:**

- Missing Concepts: Dye_Particle_Room, Temperature_Decrease, and Slower_Motion

**modification Guidance (Next Sketch Revisions by Visual Hint):**

- Water Particle Room (understand): Add markup to highlight water particles/molecules on the first block.

**Reasoning Gaps Detected In:**

- Dye_Particle_Room, Temperature_Decrease, and Slower_Motion

Figure 3: Cognitive alignment score and feedback for the perceived SRG (See Figure 2) generated by Agent 3 after similarity score calculations.

**Agent 4: Feedback Generator and Sketch Modification.**    Agent 4 in SKETCHMIND initiates a feedback loop when the similarity score $S(G_s, G_o)$ falls below a threshold $\tau$ (pre-defined to determine sketch proficiency). In our case, the curriculum expert defined three levels of proficiencies (Beginning, Developing, and Proficient) as can be seen in Figure 1b. $\tau$ value is carefully calculated for each level and used by the agent. The agent identifies missing or misaligned nodes and edges, and traces each to its expected Bloom level. Using a trained reverse mapping $\phi: (v, e) \rightarrow$ VisualHint, Agent 4 generates cognitively aligned suggestions. For instance, if a student omits a causal interaction labeled ANALYZE, the system may overlay an arrow with a textual prompt like "What causes this effect?". Below is the step-wise procedure of the sketch revision loop: Given $G_o$ and $x^{(0)}$:

1. Compute $G_s^{(t)} = f_{\text{SRG}}(x^{(t)})$.

2. If $S(G_s^{(t)}, G_o) \geq \tau$, exit loop.

3. Identify deficient elements:

$$\Delta^{(t)} = \{v \in V_s \mid v \notin \text{aligned nodes}\} \cup \left\{e \in E_s \mid e \notin E_o^{(t)}\right\}.$$

4. Use reverse mapping $\phi : (v, e) \rightarrow$ visual hint generated by Agent 1 to provide visual suggestions $H^{(t)}$.

5. Render $H^{(t)}$ on sketch canvas and generate Python code to modify the canvas and run locally.

6. Modify the canvas with updated overlay and prompting student to revise $x^{(t)} \rightarrow x^{(t+1)}$.

7. Repeat until $S \geq \tau$ or maximum iterations $T_{\max}$ reached.

This sketch revision loop directly scaffolds the student toward higher-order cognitive tasks, iterating until the revised sketch meets conceptual fidelity (Modified SRG with additional node and updated sketch using the Python toolkit for the sample image given in Fig. 4). SKETCHMIND is designed to adaptively scaffold students across diverse ability levels. If a sketch yields an incoherent or low-complexity SRG (e.g., all nodes at REMEMBER), Agent 4 shifts focus to guided reconstruction, layering in increasingly complex prompts. The evaluator distinguishes perceptual errors from conceptual ones by analyzing visual evidence $\lambda$, ensuring the feedback remains diagnostic rather than punitive. By explicitly modeling Bloom's cognitive hierarchy at each stage of analysis—from rubric parsing to feedback generation SKETCHMIND transforms sketch-based assessment into a learning-oriented, interpretable, and cognitively aligned process. SKETCHMIND positions sketching not just as a representational task but as an active, assessable pathway for scientific reasoning and growth.

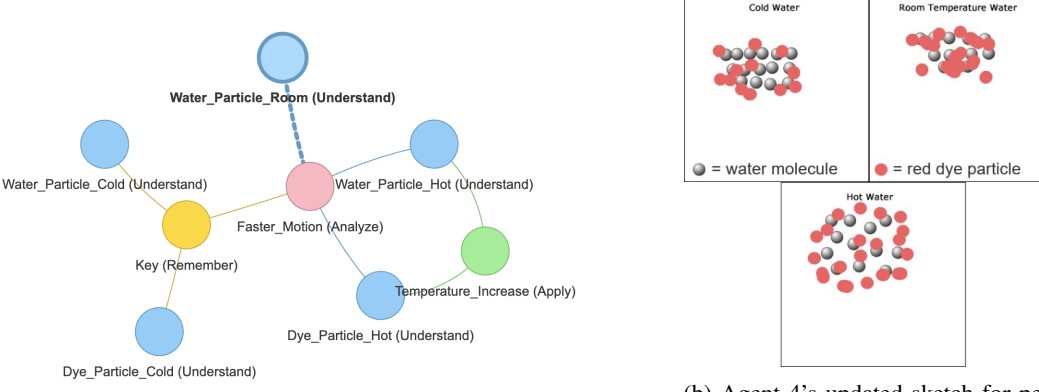

(a) Modified `SRG` with Feedback from Agent 4

(b) Agent 4's updated sketch for perceived feedback

Figure 4: Sample modified `SRG` based on the given score from Agent 3 and feedback from Agent 4 and updated sketch with embedded python toolkit.

## 4 Experimental Setup

We evaluate SKETCHMIND using two distinct model configurations to comprehensively examine its performance and versatility across closed-source and open-source MLLM frameworks. In Configuration 1, we utilize GPT-4o, GPT-4.1, GPT-4.1-nano, o3, and o4-mini (all latest GPT models with different reasoning capabilities for comprehensive evaluation) accessed via the OpenAI API. This setup leverages black-box prompting tailored explicitly to encourage structured JSON output and support multimodal reasoning. Configuration 2 deploys open-source MLLMs, specifically the INT4-quantized Llama-4 Maverick and INT8-quantized Scout (400B and 109B parameters respectively, with 17B active parameters), running locally on four NVIDIA H100 GPUs, facilitated by Hugging Face Transformers (version 4.39+) and PyTorch (version 2.1+). We keep the same model for each agent to maintain the model performance consistency, for example, if targeted GPT-4o, then each agent uses GPT-4o and mainatin local chat session.

For closed-source configurations, inference requests are systematically routed through the OpenAI API, meticulously logging timestamps, model versions, and full prompt-response interactions to ensure transparency and reproducibility. Open-source models are executed on-premises, utilizing four H100 GPUs with CUDA 12.1 and cuDNN 8.9, loading quantized model weights from the Hugging Face Model Hub, and multimodal data inputs are managed by the `Llama4Processor`.

Both configurations operate with a multi-agent pipeline detailed in Section 3, where all the agents run in sequence. Agents 1, 2, and 4 incorporate specifically designed prompt templates optimized for effective instruction-following, structured output enforcement, and multimodal reasoning integration. Agent 3 functions as a deterministic and model-agnostic component, focusing exclusively on graph comparison tasks. Comprehensive documentation of prompt templates is available in supplementary materials to enable precise experimental replication.

**Dataset and Evaluation.** Despite growing interest in applying machine learning to educational domains, there remains a significant lack of publicly available, high-quality datasets that capture student-generated visual reasoning, particularly scientific sketches. To our knowledge, no large-scale, general-purpose benchmark exists that includes both raw student-drawn diagrams and structured expert annotations for meaningful semantic evaluation.

Given this gap, we base our study on a rigorously developed dataset originally introduced by Zhai et al. [30], which has since become one of the most widely recognized resources for evaluating automated reasoning over student-generated scientific models. This dataset, adapted from the NGSA (Next Generation Science Assessment) initiative [10], aligns closely with the NGSS framework [26] and has been used by AIED researchers to assess students' conceptual understanding through multimodal evidence [16, 15]. Comprehensive details about dataset selection rationale and statistics are provided in Appendix B. To ensure pedagogical validity, we conducted a structured human evaluation of model-generated feedback. Four domain-expert educators, each with graduate-level

training in science education, independently assessed the pedagogical quality of system responses. Each rater evaluated a stratified random sample of 890 student-generated sketches (25% of the dataset), ensuring balanced representation across models (GPT-4o, GPT-4.1, O3), grade levels, and science task types.

All evaluators participated in a calibration phase: they jointly annotated and discussed 10 representative examples to align rubric interpretation, and then independently scored the same 10 sketches. Inter-rater reliability, measured using Quadratic Weighted Kappa, reached $\kappa = 0.83$, which is consistent with benchmarks for expert judgment in science assessment [11]. For the main study, each response was rated by at least two experts in a double-blind manner.

Evaluation guidelines were based on an extended rubric adapted from Zhai et al. [30] and tailored for formative science assessment. Experts rated each response along three dimensions: **1) Clarity:** Whether the textual and visual feedback was understandable and actionable. **2) Conceptual Accuracy:** Whether suggested modifications and explanations correctly reflected the target scientific concepts. And **3)Instructional Value:** The feedback's potential to promote student learning and progression along Bloom's taxonomy. Each dimension was scored on a 5-point ordinal scale (1–5). Consistency across raters was supported through the initial calibration process, and ratings were conducted under double-blind conditions to mitigate bias. Framework–human agreement rates, along with detailed rubric codebooks and annotated examples, are reported in Section 4 and the supplementary material.

**Implementation Details.** The `SRG` construction pipeline in SKETCHMIND utilizes a shared `SRGBuilder` class, which efficiently constructs, validates, and caches graphs, significantly reducing computational overhead and cost during repeated evaluations. Sketch adequacy is determined by a similarity threshold ($\tau = 0.75$), with dynamic generation of visual hints guided by a reverse mapping ($\phi$) embedded within Agent 1's implementation. We calculate the sketch prediction accuracy for each assessment item by comparing with human-expert annotated proficiencies as (Sum of correctly predicted samples across each proficiency level/Total samples) and average for all items as (Sum of all item's accuracies/Total number of items). We have evaluated the performance of SKECTHMIND by decomposing it into combination of target model with proposed `SRG`. This decomposition can help us understand the impact of target model and proposed `SRG` to determination best possible combination for SKETCHMIND. Detailed evaluation scripts and additional implementation specifics are provided comprehensively in Appendix A.

## 5 Results

Table 1 presents item-wise and macro-average accuracy across a range of MLLMs, both with and without `SRG` integration. The results consistently demonstrate that incorporating `SRG` supervision significantly improves performance across all models and items. For instance, GPT-4o, shows a substantial accuracy increase from 47.7% to 76.5% on Item H4-1, a relative gain of nearly 30 percentage points. Averaged across all items, GPT-4o benefits from an improvement of approximately 21.4%. Even state-of-the-art models such as GPT-4.1 show meaningful accuracy gains ranging from 11.0% to 15.4% when `SRG` guidance is applied, with performance increasing from 74.2% to 89.6% on Item R1-1 and from 73.1% to 87.2% on Item H4-1. Hence, SKETCHMIND works best with GPT-4.1 integrated with `SRG`.

Models with lower baseline performance, such as LLaMA 4 Maverick and Scout, experience even greater relative improvements. LLaMA 4 Maverick, for example, improves by 29.7% on Item J2-1 and achieves up to 26.0% gains on Item H4-1, suggesting that structured supervision via `SRG`s can dramatically enhance reasoning capabilities in non-reasoning models for open-source SKETCHMIND.

**Why Multi-Agent Framework?** All the above-mentioned results are performed with the multi-agent framework proposed for SKETCHMIND, but here comes the question: why not a single agent? To answer that and assess the impact of modularization inherent in the multi-agent framework for reasoning tasks, we conducted an ablation study comparing a unified single-agent framework with our proposed multi-agent pipeline. Each setting was evaluated with and without `SRG` supervision using two strong backbone models: GPT-4o and GPT-4.1.

The results in Table 2 demonstrate that modularizing the reasoning process via a multi-agent framework consistently improves SKETCHMIND's performance over the single-agent baseline across

Table 1: Item-wise accuracy (%) across models with and without SRG integration for SKETCHMIND.

| Model | R1-1 | J2-1 | M3-1 | H4-1 | H5-1 | J6-1 | Average |
|---|---|---|---|---|---|---|---|
| GPT-4o | 63.2 | 58.4 | 53.5 | 47.7 | 52.3 | 58.6 | 55.6 |
| + SRG | **78.5** | **77.4** | **75.8** | **76.5** | **74.7** | **79.1** | **77.1** |
| *Gain* | *+15.3* | *+19.0* | *+22.3* | *+28.8* | *+22.4* | *+20.5* | *+21.4* |
| GPT-4.1 | 74.2 | 78.5 | 77.4 | 73.1 | 79.6 | 81.5 | 77.4 |
| + SRG | **89.6** | **91.6** | **88.4** | **87.2** | **91.7** | **92.6** | **90.2** |
| *Gain* | *+15.4* | *+13.1* | *+11.0* | *+14.1* | *+12.1* | *+11.1* | *+12.8* |
| GPT-4.1-nano | 62.5 | 61.3 | 59.6 | 57.3 | 63.8 | 67.5 | 62.0 |
| + SRG | **73.7** | **72.6** | **70.4** | **68.7** | **78.5** | **79.3** | **73.9** |
| *Gain* | *+11.2* | *+11.3* | *+10.8* | *+11.4* | *+14.7* | *+11.8* | *+11.9* |
| O3 | 75.2 | 79.5 | 76.4 | 75.3 | 77.5 | 79.4 | 77.2 |
| + SRG | **89.5** | **91.1** | **87.3** | **86.4** | **89.6** | **90.3** | 89.0 |
| *Gain* | *+14.3* | *+11.6* | *+10.9* | *+11.1* | *+12.1* | *+10.9* | *+11.8* |
| O4-mini | 71.4 | 75.3 | 73.3 | 69.2 | 74.6 | 76.1 | 73.3 |
| + SRG | **79.5** | **81.5** | **79.4** | **77.6** | **82.8** | **83.9** | **80.8** |
| *Gain* | *+8.1* | *+6.2* | *+6.1* | *+8.4* | *+8.2* | *+7.8* | *+7.5* |
| LLaMA 4 Scout | 48.6 | 43.2 | 39.5 | 38.4 | 45.6 | 47.5 | 43.8 |
| + SRG | **63.8** | **69.3** | **59.6** | **61.7** | **67.5** | **66.8** | **64.8** |
| *Gain* | *+15.2* | *+26.1* | *+20.1* | *+23.3* | *+21.9* | *+19.3* | *+21.0* |
| LLaMA 4 Maverick | 53.4 | 49.7 | 44.5 | 42.7 | 46.8 | 49.6 | 47.8 |
| + SRG | **77.3** | **79.4** | **63.6** | **68.7** | **71.8** | **73.5** | **72.4** |
| *Gain* | *+23.9* | *+29.7* | *+19.1* | *+26.0* | *+25.0* | *+23.9* | *+24.9* |

all six items. Without SRG integration, GPT-4o's accuracy increases from 50.1% (single-agent) to 55.6% (multi-agent), while GPT-4.1 improves from 62.9% to 77.4%, indicating that decomposing tasks into specialized agents enables more structured, context-aware reasoning even without explicit graph guidance. This performance gap widens with SRG supervision: GPT-4o's accuracy rises from 69.5% to 77.1% and GPT-4.1 from 82.8% to 90.2%, with item-wise gains ranging from 5.2% to 13.4%. Notably, Item H4-1 (Hot Shower Effect), which demands the highest Bloom's taxonomy level (Create), sees accuracy climb from 63.2% to 76.5% for GPT-4o and from 79.3% to 87.2% for GPT-4.1 when switching to multi-agent reasoning with SRG, confirming that modular agents are better equipped to leverage structured semantic guidance. These results highlight that a multi-agent architecture, where reasoning responsibilities are explicitly segmented and coordinated, facilitates more robust and interpretable scientific reasoning.

Table 2: Ablation study comparing single-agent and multi-agent frameworks for SKETCHMIND (accuracy in %)

| Model configurations | R1-1 | J2-1 | M3-1 | H4-1 | H5-1 | J6-1 | Average |
|---|---|---|---|---|---|---|---|
| GPT-4o (Single Agent w/o SRG) | 56.3 | 52.4 | 49.3 | 43.1 | 48.3 | 51.2 | 50.1 |
| GPT-4o (Multi-Agent w/o SRG) | 63.2 | 58.4 | 53.5 | 47.7 | 52.3 | 58.6 | 55.6 |
| GPT-4o (Single Agent w/ SRG) | 73.5 | 69.6 | 68.4 | 63.2 | 68.8 | 74.3 | 69.5 |
| GPT-4o (Multi-Agent w/ SRG) | **78.5** | **77.4** | **75.8** | **76.5** | **74.7** | **79.1** | **77.1** |
| GPT-4.1 (Single Agent w/o SRG) | 69.6 | 61.3 | 59.2 | 57.1 | 58.8 | 71.2 | 62.9 |
| GPT-4.1 (Multi-Agent w/o SRG) | 74.2 | 78.5 | 77.4 | 73.1 | 79.6 | 81.5 | 77.4 |
| GPT-4.1 (Single Agent w/ SRG) | 84.4 | 81.2 | 83.6 | 79.3 | 82.7 | 85.3 | 82.8 |
| GPT-4.1 (Multi-Agent w/ SRG) | **89.6** | **91.6** | **88.4** | **87.2** | **91.7** | **92.6** | **90.2** |

**Feedback and Sketch Modification Evaluation.** To assess the pedagogical quality and usefulness of SKETCHMIND's generated sketches and feedback, we conducted a human evaluation study focusing on Agent 4, the component responsible for modification of visual representations and

providing formative feedback. Expert evaluators, comprising experienced science educators, rated the outputs across six science items based on their clarity, conceptual accuracy, and instructional value.

As shown in Table 3, GPT-4.1 achieved the highest average rating (4.1), followed closely by o3 (4.0) and LLaMA 4 Maverick (3.5). Evaluators consistently noted that sketches generated using GPT-4.1 were pedagogically sound, well-aligned with scientific principles, and accompanied by feedback that could directly support improved student learning. Lower-performing models, such as GPT-4o and LLaMA 4 Scout, received average ratings of 2.3 and 2.5, respectively, often due to missing or vague concepts and less actionable feedback. The evaluators emphasized that when integrated with high-performing language models like GPT-4.1, SKETCHMIND has the potential to significantly improve the quality of student-generated scientific sketches through guided modification and tailored feedback.

Table 3: Human Evaluation Ratings of Feedback and Sketch Modification (1 = Poor, 5 = Excellent).

| Model | R1-1 | J2-1 | M3-1 | H4-1 | H5-1 | J6-1 | Average |
|---|---|---|---|---|---|---|---|
| GPT-4o | 2.5 | 2.0 | 2.5 | 2.5 | 2.5 | 2.0 | 2.3 |
| GPT-4.1 | **4.5** | **4.0** | **3.5** | **3.5** | **4.5** | **4.5** | **4.1** |
| GPT-4.1-nano | 3.0 | 3.0 | 3.5 | 2.5 | 3.0 | 3.5 | 3.1 |
| O3 | 4.5 | 4.0 | 4.0 | 4.0 | 3.5 | 4.0 | 4.0 |
| O4-mini | 4.0 | 3.5 | 3.0 | 3.0 | 3.5 | 3.0 | 3.3 |
| LLaMA 4 Scout | 3.0 | 2.5 | 2.0 | 2.5 | 2.0 | 3.0 | 2.5 |
| LLaMA 4 Maverick | 3.5 | 4.0 | 3.5 | 3.5 | 3.0 | 3.5 | 3.5 |

# 6   Conclusion

In this work, we introduced SKETCHMIND, a cognitively grounded multi-agent system for assessing and improving student-generated scientific sketches. By leveraging SRGs annotated with Bloom's taxonomy, SKETCHMIND enables interpretable evaluation, pedagogically aligned feedback, and iterative modification of higher-quality visual explanations. Empirical results across six NGSS-aligned items show that SKETCHMIND (model + SRG) substantially outperforms both MLLM baselines without SRG and single-agent pipelines, achieving up to 90.2% average accuracy with GPT-4.1 and receiving high expert ratings of (4.1) for feedback quality. Overall, each MLLM shows significant improvement for sketch reasoning and prediction with the proposed SRG integration which highlights the significance of structural reasoning for scientific sketch evaluations. SKETCHMIND bridges the gap between visual AI reasoning and education by embedding cognitive theories directly into the assessment pipeline.

**Limitations and future directions.**   Despite promising results, several limitations warrant attention. First, while the multi-agent system is modular, inter-agent coordination is static and predefined, future work could explore dynamic planning strategies using large language model (LLM) controllers or reinforcement learning, as demonstrated in recent advances in task decomposition for multi-agent collaboration [7, 9]. Second, our findings relies on overal sketch prediction for given proficieicny level; however, SRG-level evaluation may provide in-depth analysis of predictions. Future work could involve experts to create manual SRGs for in-depth analysis. Lastly, Incorporating student behavioral data (e.g., stroke sequence or eye tracking) into the SRG modeling pipeline may also further enhance alignment with cognitive engagement signals [14].

## Acknowledgments and Disclosure of Funding

The research reported here was supported by the Institute of Education Sciences, U.S. Department of Education, through Grant R305C240010 (PI Zhai). The opinions expressed are those of the authors and do not represent views of the Institute or the U.S. Department of Education.

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

# Appendices

# A    Prompts and Code

**Prompts for Cognitive Sketch Assessment Agents**   This section details the prompt structures used to interact with the MLLMS for agents 1, 2, and 4 in our cognitive sketch assessment pipeline. The prompts are designed to elicit structured JSON outputs and guide the MLLMs through their respective tasks, including SRG generation, sketch analysis, and visual feedback generation. For clarity and visualization in this document, prompts containing Python class definitions or multi-line JSON examples are presented within code-like blocks; in our actual implementation, these are passed as standard text strings to the MLLM. The placeholders like `[QUESTION_TEXT]`, `[RUBRIC_TEXT]`, etc., are dynamically filled at runtime.

**Agent-1:**   This agent is responsible for interpreting the educational question and rubric, along with golden standard sketches, to generate a reference SRG and a mapping of concepts to visual drawing hints.

---

**System Prompt**

You are Agent 1: Rubric Parser in a cognitive sketch evaluation system.
Your job is to construct a Bloom-aligned Sketch Reasoning Graph (SRG) based on the rubric, question, and example sketches.
Use the SRGBuilder class `[SRGBuilder_Class_String_Placeholder]` to:
1. Extract semantic concepts relevant to the given question and rubric.
2. Label them with Bloom levels (Remember, Understand, Apply, Analyze, Evaluate, Create).
3. Define directed edges that capture causal or logical relationships.
4. Use example sketches to identify additional visual concepts and reinforce what is expected.
5. Validate the graph for connectivity (There should be a path from each node to other nodes) and Bloom level ordering.
6. Provide a reverse mapping from concepts to drawing hints using Bloom's taxonomy.
Return JSON in this format:

```
{
  "srg": {"nodes": [{"label":"...", "bloom_level":"..."}], "edges":
  ↪  [{"from":"...", "to":"..."}]},
  "reverse_mapping": {"concept_name": "visual_hint"}
}
```

Be consistent with node naming across examples. Be thorough but not redundant.

---

Note: `[SRGBuilder_Class_String_Placeholder]` is replaced by the full string definition of the `SRGBuilder` class as shown below:

```python
class SRGBuilder:
    BLOOM_ORDER = ["Remember", "Understand", "Apply", "Analyze", "Evaluate",
    ↪  "Create"]

    def __init__(self, question: str, rubric: str):
        self.question = question
        self.rubric = rubric
        self.nodes: List[Tuple[str, str]] = []   # (concept, Bloom level)
        self.edges: List[Tuple[str, str]] = []   # (from, to)

    def add_node(self, concept: str, bloom_level: str):
        assert bloom_level in self.BLOOM_ORDER, f"Invalid Bloom level:
        ↪  {bloom_level}"
        self.nodes.append((concept, bloom_level))

    def add_edge(self, source: str, target: str):
        self.edges.append((source, target))

    def build_graph(self) -> Dict[str, List[Tuple[str, str]]]:
        return {"nodes": self.nodes, "edges": self.edges}
```

```python
def validate_graph(self) -> bool:
    node_names = {n[0] for n in self.nodes}
    valid_edges = all(u in node_names and v in node_names for u, v in
    ↪  self.edges)

    # Check for connectivity
    G = nx.DiGraph()
    G.add_edges_from(self.edges)
    connected = nx.is_weakly_connected(G) if G.number_of_nodes() > 0 else
    ↪  False

    # Ensure Bloom level ordering is respected
    bloom_levels = [self.BLOOM_ORDER.index(level) for _, level in self.nodes]
    ordered = bloom_levels == sorted(bloom_levels)

    return valid_edges and connected and ordered
```

## Initial User Prompt Sequence

The initial interaction with Agent-1 involves a sequence of user messages:

1. **Text Input:**

   ```
   Question: [QUESTION_TEXT]
   Rubric: [RUBRIC_TEXT]
   Please analyze the following golden standard sketches for
   guidance.
   ```

2. For each of the 3 golden standard images provided:
   - **Image Input:** [GOLDEN_STANDARD_IMAGE_N_BASE64]
   - **Associated Text:** "This is a golden standard sketch."

**Agent-2:** This agent analyzes the student's sketch, using a reference SRG (from Agent-1) as a template, to identify concepts and relationships present in the sketch.

## System Prompt

You are Agent 2: Sketch Parser in a cognitive sketch evaluation system.
You receive a student's sketch and analyze its contents to construct a Sketch Reasoning Graph (SRG).
Your SRG output must use the same node labels and edge labels as in the reference SRG if possible.
Only include a node or edge if it is visibly present or clearly inferable.
Instructions:
1. Use the reference SRG node and edge names as a template.
2. Detect which concepts and relationships from the reference are actually present in the sketch.
3. Label each node with the Bloom level shown by the sketch.
4. Return only valid components and their Bloom levels.
Return JSON in the format:

```json
{
  "srg": {
    "nodes": [{"label": "...", "bloom_level": "..."}],
    "edges": [{"from": "...", "to": "..."}]
  }
}
```

Be strict and do not assume missing content. Match names exactly where applicable.

**Initial User Prompt Sequence**

The initial interaction with Agent 2 involves:

1. **Text Input (Reference SRG Information):**

   ```
   Reference SRG Node Labels:
   - [Label1]
   - [Label2]
   ...

   Reference SRG Edges:
   - [Source1] -> [Target1]
   - [Source2] -> [Target2]
   ...
   ```

2. **Image Input (Student's Sketch):** `[STUDENT_SKETCH_IMAGE_BASE64]`

**Prompt Variations for LLaMA-4 (Maverick & Scout)** While the core structure of all prompts remained identical, for the LLaMA-4 variants we applied only these minimal changes to get the same structured JSON replies as our GPT-based agents :

**Agent-1 (LLaMA-4 Variant):** Only the bold content in the following lines differ from the GPT-based prompts above:

**System Prompt**

Use the SRGBuilder class `[SRGBuilder_Class_String_Placeholder]` to:
Follow these steps:
1. Extract **concise** semantic concepts relevant to the question and rubric.
2. Label **each concept** with **its corresponding** Bloom's taxonomy level: Remember, Understand, Apply, Analyze, Evaluate, or Create.
3. Define directed edges that capture causal or logical relationships **between these concepts**.
4. **Incorporate insights from** example sketches to identify additional visual concepts and reinforce expectations.
5. ...
6. ...
**Your output must strictly adhere to the following JSON format**:

```
{
  "srg": {"nodes": [{"label":"...", "bloom_level":"..."}], "edges":
  ↪  [{"from":"...", "to":"..."}]},
  "reverse_mapping": {"concept_name": "visual_hint"}
}
```

**Use double quotes for all keys and string values. Do not include any explanations or additional text outside the JSON structure.**
...

**Initial User Prompt Sequence**

```
...
...
Please analyze the above information and generate the Sketch Reasoning
Graph (SRG) and reverse mapping as per the system instructions and
Please analyze the following golden standard sketches for guidance
```

```
...
```

**Agent-2 (LLaMA-4 Variant):** Only the bold content in the following lines differ from the GPT-based prompts above:

### System Prompt

...

...

Return JSON in the format **(double quotes only)**:

...

**Use double quotes for all keys and string values. Do not include any explanations or additional text outside the JSON structure.**

**Agent-4:** While Agent 4 performs several functions, its primary MLLM interaction for generating new content involves creating Python code for visual overlays on the student's sketch. Agent-3 is deterministic and does not use an MLLM.

### System Prompt

You are Agent-4: That returns Python code to overlay visual sketch hints on images. Before generating the code, understand the given image on which this overlay will apply, and carefully position objects appropriately. Do not just randomly place the overlay on the image in the code.

### Initial User Prompt Sequence

This prompt is sent along with the student's sketch image.

1. **Text Input:**

```
Generate a Python function using PIL to visually overlay a hint
onto a student sketch.
The original image is [IMAGE_WIDTH]px wide and [IMAGE_HEIGHT]px
tall.
Concept: [CONCEPT_NAME_FOR_HINT]
Hint: [GENERATED_VISUAL_HINT_TEXT]
Return only the function named
`overlay_hint(image: Image.Image) -> Image.Image`.
```

2. **Image Input (Student's Sketch):** [STUDENT_SKETCH_IMAGE_BASE64]

**Agent-3 Core Evaluation Logic.** The following code implements the similarity computation, classification, and feedback signal extraction used by Agent 3.

```python
class Agent3:
    def run(self, reference_srg, student_srg):
        ref = SRGBuilder("", "")
        stu = SRGBuilder("", "")
        ref.nodes, ref.edges = reference_srg['nodes'], reference_srg['edges']
        stu.nodes, stu.edges = student_srg['nodes'], student_srg['edges']

        score = stu.compute_similarity(reference_srg)

        missing_nodes = [n for n in reference_srg['nodes'] if n not in
        ↪   student_srg['nodes']]
```

```python
        missing_edges = [e for e in reference_srg['edges'] if e not in
        ↪   student_srg['edges']]
        irrelevant_nodes = [n for n in student_srg['nodes'] if n not in
        ↪   reference_srg['nodes']]
        irrelevant_edges = [e for e in student_srg['edges'] if e not in
        ↪   reference_srg['edges']]

        # Compare Bloom level expectations
        bloom_discrepancies = []
        ref_node_dict = dict(reference_srg['nodes'])
        for concept, level in student_srg['nodes']:
            if concept in ref_node_dict and level != ref_node_dict[concept]:
                bloom_discrepancies.append({
                    "concept": concept,
                    "expected": ref_node_dict[concept],
                    "observed": level
                })

        # Classify sketch based on similarity score
        if score >= SCORE_THRESHOLD:
            label = "Proficient"
        elif score >= 0.5:
            label = "Developing"
        else:
            label = "Beginning"

        # Rank missing nodes by Bloom level
        priority_fix = sorted(missing_nodes, key=lambda x:
        ↪   SRGBuilder.BLOOM_ORDER.index(x[1]))

        # Detect gaps in reasoning flow
        expected_sources = {src for src, _ in reference_srg['edges']}
        actual_sources = {src for src, _ in student_srg['edges']}
        conceptual_gaps = list(expected_sources - actual_sources)

        return {
            "similarity_score": round(score, 3),
            "classification": label,
            "missing_nodes": missing_nodes,
            "missing_edges": missing_edges,
            "irrelevant_nodes": irrelevant_nodes,
            "irrelevant_edges": irrelevant_edges,
            "bloom_discrepancies": bloom_discrepancies,
            "priority_fix": priority_fix,
            "conceptual_gaps": conceptual_gaps
        }
```

## B  Dataset Statistics

We selected this dataset for three primary reasons: 1) *Relevance and Authenticity:* It contains real, open-ended student responses in the form of scientific sketches, reflecting a range of cognitive and conceptual models that go beyond synthetic or constrained data sources. This authenticity is essential for evaluating models that aim to interpret or provide feedback on student thinking. 2) *Expert Annotation and Interpretability:* Each assessment item is supported by a rubric and three gold-standard (SRGs), representing Beginning, Developing, and Proficient levels. These gold standards were constructed by domain experts, enabling supervised learning and interpretability in model predictions. 3) *Domain Breadth and Volume:* comprises six items across physical sciences spaning from physical chemistry, heat transfer, and thermodynamics, the dataset contains over 3,500 curated student sketches. Each assessment item is associated with a specific highest order of Bloom's level as

each assessment is designed for comprehensive sketch evaluation. Each response has been cleaned to remove noise and non-sensical content, making it suitable for reliable evaluation. Further, each item is supported by an expert-authored rubric and three corresponding gold-standard (SRGs), representing Beginning, Developing, and Proficient levels of performance.

The dataset consists of student drawn scientific sketches and golden-standard sketches drawn by teacher. A "scientific sketch" is a visual representation intended to illustrate a scientific concept, process, or phenomenon. For example, for thermodynamics concept, the scientific sketch shows heat source, its transition phases, arrows may represent the flow of heat, and object/medium transferring the heat. Teacher-drawn sketches serve as gold-standard reference sketches, explicitly containing essential visual components necessary to fully represent the target scientific concept (e.g., arrows clearly indicating heat transfer direction in thermodynamics). In contrast, student-drawn sketches often reflect incomplete, varied, or erroneous understanding, exhibiting substantial diversity and ambiguity. We explicitly distinguish these categories in our data annotation by using teacher sketches as gold-standard "reference SRGs" and grounding student sketches pedagogically to Bloom's cognitive levels (See Fig. 1. We also have included the representative examples, illustrating these differences in Fig. 5.

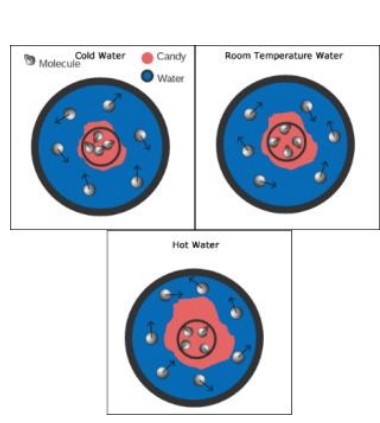

(a) Teacher Drawn golden-standard sketch for the repsentative example shown in Fig. 1 reaching Level 6

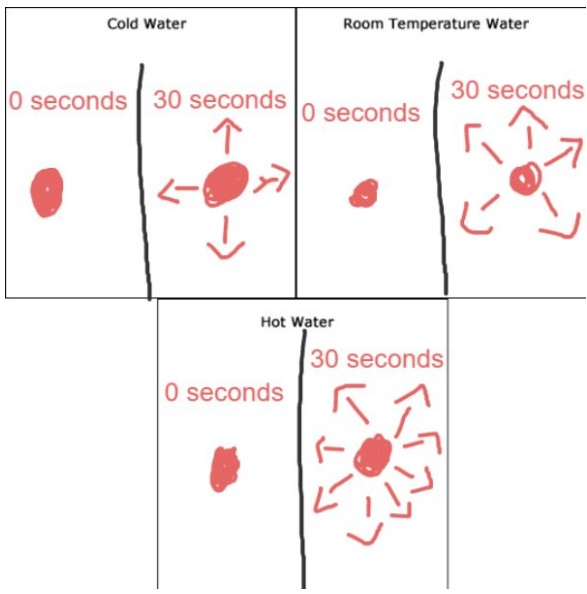

(b) A student drawn scientific sketch (Barely reaching the Understanding Level of Bloom's Teaxonomy)

Figure 5: Sample sketches drawn by teacher and student(a) Teacher's drawn sketch (Level 6, (b) Student drawn sketch at Level 1

Table 4: Summary of Assessment Items and Student Response Distribution based on the Proficiency Level.

| Item# | Name | Bloom's Level | Total Samples | Beginning | Developing | Proficient |
|-------|------|---------------|---------------|-----------|------------|------------|
| R1-1 | Red dye diffusion | 4 (Analyze) | 476 | 194 | 205 | 77 |
| J2-1 | Jane's inflated ball | 5 (Evaluate) | 538 | 177 | 288 | 73 |
| M3-1 | Melting Butter | 5 (Evaluate) | 520 | 155 | 266 | 99 |
| H4-1 | Hot Shower Effect | 6 (Create) | 772 | 494 | 107 | 171 |
| H5-1 | Heated Cup of Water | 5 (Evaluate) | 453 | 61 | 262 | 130 |
| J6-1 | Jennifer's Teapot | 4 (Analyze) | 816 | 390 | 271 | 155 |

The Table. 4 summarizes each item, its domain, the distribution of student responses across the three proficiency levels, and highest order of Bloom's level. Evaluation criteria for model quality include: structural validity and semantic alignment via accurate node and edge labeling consistent with Bloom's taxonomy in term of accuracy for final predictions based on SRGs, and feedback quality

as rated by domain experts on a 5-point Likert scale (1 = poor, 5 = excellent), assessing the clarity and instructional value of generated visual hints and textual explanations.

## C   Costs

Table 5: Cost and Token Usage Per Sample across all Agents for each Model.

| Item# | Model | Input Tokens/sample | Output Tokens/sample | Cost per sample | E2E Latency |
|-------|-------|---------------------|----------------------|-----------------|-------------|
| R1-1 | GPT-4o | ∼5,105 | ∼1491 | $0.0479 | 9.48s |
| R1-1 | GPT-4.1 | ∼5,105 | ∼1491 | $0.0221 | 20.35s |
| R1-1 | GPT-4.1 nano | ∼5,105 | ∼1491 | $0.0011 | 17.16s |
| R1-1 | O3 | ∼5,105 | ∼1491 | $0.1107 | 51.62s |
| R1-1 | O4-mini | ∼5,105 | ∼1491 | $0.0122 | 169.79s |
| R1-1 | Llama-4 Scout | ∼5,170 | ∼1351 | Local Inference | 13.30s |
| R1-1 | Llama-4 Maverick | ∼5,170 | ∼1351 | Local Inference | 13.82s |

## D   Additional Evaluation

## E   Impact Statement

SKETCHMIND advances AI in Education by introducing a cognitively grounded, multi-agent system for evaluating student-drawn scientific sketches. By combining semantic reasoning with Bloom's Taxonomy, it enables interpretable, standards-aligned feedback that supports formative assessment. This approach benefits science educators by automating consistent, high-quality evaluation and guiding students through conceptually meaningful revisions. For researchers, SKETCHMIND provides a reproducible framework and dataset for studying multimodal reasoning, cognitive alignment, and agentic learning systems—paving the way for more transparent and pedagogically sound AI tools in STEM education.

