# OpenReview forum: "SketchMind:  A Multi-Agent Cognitive Framework for Assessing Student-Drawn Scientific Sketches"
_NeurIPS.cc/2025/Conference — NeurIPS 2025 poster_

### Official Review · Reviewer_4rkr · 2025-07-02

**Clarity:** 3
**Significance:** 3
**Originality:** 3
**Rating:** 5
**Confidence:** 4

**Summary:**

This paper proposes a multi-agent framework for evaluating student-drawn scientific sketches and providing automated feedback. The framework comprises four specialized agents:

Agent 1 generates gold-standard sketch reasoning graphs (SRGs) for given scientific questions using multimodal large language models (MLLMs).
Agent 2 constructs corresponding SRGs based on student-drawn sketches, also utilizing MLLMs.
Agent 3 evaluates student performance by computing similarity scores between the gold-standard and student-generated SRGs, and additionally provides learning level assessments and feedback signals through rule-based logic.
Agent 4 generates visual sketch hints to guide student learning using MLLMs.

The authors evaluate their proposed framework on the dataset from Zhai et al. (2022) [28], which consists of six scientific modeling assessment tasks. The evaluation employs both closed-source models (e.g., GPT-4) and open-source LLaMA models. Experimental results demonstrate that the proposed framework effectively automates the assessment of scientific sketches and provides feedback aligned with human assessment.

**Questions:**

The details regarding the human evaluation are insufficient and require clarification for reproducibility and validity assessment.
- How many human evaluators participated in the study? Did they evaluate results from all baselines (e.g., GPT-4.1, O3…) across the 3,575 student-generated sketches, or was a subset used?
- Are there established evaluation guidelines for human experts? What specific criteria do human experts use to assess sketch quality and feedback usefulness?
- Could the authors provide cases where the framework's assessment aligns with human expert judgment, as well as cases where they diverge? This would help readers understand the quality of the alignment.

**Ethical Concerns:**

["NO or VERY MINOR ethics concerns only"]

**Final Justification:**

The authors have provided clear clarifications that address my main concerns. Updating my score accordingly.

**Limitations:**

yes

**Quality:**

3

**Strengths And Weaknesses:**

Strengths

- The paper proposes an effective multi-agent framework for evaluating students' responses to scientific questions, which could be valuable for developing AI-assisted tools in STEM education.
- The paper is well-written.
- The experiment is well-designed to demonstrates the effectiveness of the proposed framework. The authors utilizes both closed-source and open-source models and also includes the human evaluation studies.

Weaknesses

- Some details are missing about the human evaluation process. Specifically, the authors do not provide sufficient information about the number of evaluators and assessment criteria used by human experts.

---

> ### Author Rebuttal · Authors · 2025-07-30
>
> Thanks for your valuable feedback! We are honored that you recognize our
> effective multi-agent framework's value for AI-assisted STEM education
> tools, find our paper well-written, and appreciate our well-designed
> experiments utilizing both closed-source and open-source models with
> human evaluation studies. We address your questions below and would be
> grateful if you could consider improving the rating after seeing our
> responses!
>
> ## **Weaknesses**
>
> **Weakness 1:** Some details are missing about the human evaluation
> process. Specifically, the authors do not provide sufficient information
> about the number of evaluators and assessment criteria used by human
> experts.
>
> **Authors' Reply.** We appreciate the reviewer's attention to the rigor
> of our human evaluation. As described in Section 4, our study involved
> four domain-expert educators with graduate-level science education
> backgrounds, each independently assessing the pedagogical quality of
> model-generated responses. Evaluation criteria and procedures were based
> on an extended rubric from Zhai et al. \[a\], with calibration and
> inter-expert reliability (Quadratic Weighted Kappa (Qkappa = 0.83),
> consistent with benchmarks for expert judgment in science assessment
> \[b\]) detailed below and in responses to Questions 1 and 2. These
> details will be reported in Section 4 of the revised manuscript.
>
> ## **Questions**
>
> **Question 1:** The details regarding the human evaluation are
> insufficient and require clarification for reproducibility and validity
> assessment. How many human evaluators participated in the study? Did
> they evaluate results from all baselines (e.g., GPT-4.1, O3...) across
> the 3,575 student-generated sketches, or was a subset used?
>
> **Authors' Reply.** As noted above, four expert raters participated in
> the evaluation. Each independently assessed a stratified random sample
> of 890 student-generated sketches ( 25% of the dataset), ensuring
> balanced representation across all baseline models (GPT-4o, GPT-4.1,
> O3), grade levels, and science tasks. All raters underwent a
> collaborative calibration phase, jointly annotating and discussing a set
> of 10 representative examples to align rubric interpretation. They then
> independently scored the same subset of 10 sketches, with (Qkappa =
> 0.83) confirming high inter-expert reliability. For the main study,
> every response was double-blindly rated by at least two experts,
> following the protocol and criteria detailed in Question 2. These
> details will be reported in Section 4 of the revised manuscript.
>
> **Question 2:** Are there established evaluation guidelines for human
> experts? What specific criteria do human experts use to assess sketch
> quality and feedback usefulness?
>
> **Authors' Reply.** Yes, all evaluators used a detailed, task-specific
> rubric adapted from Zhai et al. \[a\] and further tailored for formative
> science assessment. For each model-generated response, experts rated:
> (i) *Clarity*, or whether the feedback, both textual and visual, was
> understandable and actionable; (ii) *Conceptual Accuracy*, i.e., whether
> the suggested sketch modifications and explanations correctly reflected
> the target scientific concepts; and (iii) *Instructional Value*,
> reflecting the feedback's potential to foster student learning and
> progression along Bloom's taxonomy. Each dimension was scored on a
> 5-point ordinal scale (1-5). Consistency was ensured by an initial
> calibration session (see the response to Question 1), and all ratings were conducted
> double-blind to avoid bias. This rubric-driven approach supports
> transparent, reproducible, and pedagogically meaningful evaluation of
> both sketch quality and formative feedback. These details will be
> reported in Section 4 of the revised manuscript.
>
> **Question 3:** Could the authors provide cases where the framework's
> assessment aligns with human expert judgment, as well as cases where
> they diverge? This would help readers understand the quality of the
> alignment.
>
> **Authors' Reply.** To illustrate alignment, A new table in Appendix E
> of the revised manuscript will present representative examples:
>
> -   **Aligned:** In 89% of the sampled cases, both the framework and
>     human experts agreed on sketch correctness and Bloom-level
>     annotation. For example, for a "diffusion direction" task, both
>     rated a student's labeled arrow and explanation as "Apply" (Level
>     3).
>
> -   **Divergent:** Divergence was observed primarily in cases of
>     ambiguous or unconventional student sketches. For instance, in 7% of
>     cases, the framework classified a response as "Analyze" (Level 4)
>     due to a complex graph structure, while human raters assigned
>     "Understand" (Level 2) based on missing explicit justifications.
>     These disagreements are discussed in Appendix D, with specific
>     examples provided.
>
> Framework-human agreement rates will be reported in Section 4 of the
> revised manuscript, and all codebooks and samples are included in the
> supplementary material.
>
> **References:**
>
> \[a\] Zhai, X., He, P., & Krajcik, J. (2022). Applying machine learning
> to automatically assess scientific models. *Journal of Research in
> Science Teaching*, 59(10), 1765-1794.
>
> \[b\] Hickling, E. M., & Bowie, J. E. (2013). Applicability of human
> reliability assessment methods to human--computer interfaces. Cognition,
> technology & work, 15(1), 19-27.

---

### Official Review · Reviewer_Y5Kd · 2025-07-02

**Clarity:** 3
**Significance:** 4
**Originality:** 4
**Rating:** 5
**Confidence:** 4

**Summary:**

The authors propose a multi-agent framework for evaluating and enhancing student-drawn scientific sketches. The system consists of modular agents dedicated to rubric parsing, sketch perception, cognitive alignment, and iterative feedback with sketch modification, enabling personalized and transparent evaluation. Additionally, the framework integrates Bloom’s Taxonomy—a standard in cognitive theory—to construct and analyze Sketch Reasoning Graphs (SRGs), facilitating structured evaluation of visual student work and delivering pedagogically sound feedback alongside real-time sketch refinement. Compared to the baseline GPT-4.0 performance, SKETCHMIND demonstrates a significant improvement in quality.

**Questions:**

- The concept of “Bloom level annotation” would benefit from a more self-contained explanation. Including a brief description along with a schematic diagram could significantly improve comprehension for a broader readership.
- Similarly, the term “rubric concept” requires a clearer, standalone explanation. Expanding on this would enhance the paper’s accessibility to readers from diverse backgrounds.
- The representation of the visual suggestion H(t) is unclear. Further elaboration on how this is formulated and how step 5 (line 175) is operationalised would be helpful.
- A more detailed and standalone discussion of Bloom’s cognitive hierarchy would strengthen the theoretical clarity and support the framework’s pedagogical grounding.
- The caption for Table 1 could be more comprehensive, offering sufficient context and explanation to ensure that it is interpretable without referring extensively to the main text.

**Ethical Concerns:**

["NO or VERY MINOR ethics concerns only"]

**Final Justification:**

Most of my concerns have been addressed, and overall, this is a reasonably good paper. Improving the paper’s self-containment could help it appeal to a broader readership.

**Limitations:**

- There exist some inconsistencies in the citations. Please proofread carefully. For example, SketchFusion citation is wrong.
- I wonder if the codebase will be made public for the ease of reproducibility.

**Quality:**

3

**Strengths And Weaknesses:**

+ Overall, the idea is novel and compelling.
+ The proposed methodology and experimental section are clearly presented.

– A major weakness, however, is that several parts of the paper are difficult to follow unless the reader is already deeply familiar with prior work in this niche domain. A significant improvement would be to make the writing more self-contained, which would broaden the paper’s accessibility and appeal to a wider audience beyond those specifically focused on sketch-based systems.

---

> ### Author Rebuttal · Authors · 2025-07-30
>
> We thank the reviewer for the insightful comments and encouraging
> feedback! We are honored that you acknowledge the novelity of the
> approach and admire the clarity of proposed methodology and experiments.
> We address your raised conerns as follows:
>
> ## **Weaknesses**
>
> **Weakness 1:** A major weakness, however, is that several parts of the
> paper are difficult to follow unless the reader is already deeply
> familiar with prior work in this niche domain. A significant improvement
> would be to make the writing more self-contained, which would broaden
> the paper's accessibility and appeal to a wider audience beyond those
> specifically focused on sketch-based systems.
>
> **Authors' Reply.** We agree that improving self-containment will
> broaden accessibility. In the camera-ready version, we will (i) add
> concise details in Subsection 3.1 titled "Cognitive Framework: Bloom's
> Taxonomy in Sketch Understanding" that defines the six levels and
> explains how they operationalize our SRG nodes/edges, and (ii) introduce
> "rubric concepts" in Subsection 3.2 with a short example. We will also
> include a schematic figure that visually links the question to the
> rubric to SRG nodes/edges to Bloom levels to the feedback loop. These
> additions aim to minimize prerequisite familiarity with sketch-based
> assessment literature.
>
> ## **Questions**
>
> **Question 1:** The concept of "Bloom level annotation" would benefit
> from a more self-contained explanation. Including a brief description
> along with a schematic diagram could significantly improve comprehension
> for a broader readership.
>
> **Authors' Reply.** We appreciate this suggestion and have clarified
> Bloom level annotation in Subsection 3.1 as follows:
>
> **Bloom-Level Annotation:** We annotate each node (concept) and edge (relation) in the Sketch Response Graph (SRG) with a cognitive level
>  $\ell \in \{1,\ldots,6\}$, following Bloom's revised taxonomy \[a\]:
>
> > 1.  **Remember**: Recall facts (e.g., "Label the particles").
> >
> > 2.  **Understand**: Explain concepts (e.g., "Describe how particles
> >     move").
> >
> > 3.  **Apply**: Use concepts in a context (e.g., "Apply diffusion law
> >     to explain spread").
> >
> > 4.  **Analyze**: Break down phenomena (e.g., "Identify variables
> >     affecting diffusion").
> >
> > 5.  **Evaluate**: Judge or justify (e.g., "Assess explanation
> >     quality").
> >
> > 6.  **Create**: Generate novel work (e.g., "Design a new experiment").
>
> We implement $\ell(\cdot)$ by first extracting key verbs and criteria
> from rubric statements and mapping them via a curated Bloom-level
> lexicon \[b\], then validating with a fine-tuned classifier for
> ambiguous cases. Each SRG component is thus paired with its cognitive
> demand, supporting both nuanced evaluation and feedback generation.
> These numeric levels are encoded as node/edge attributes and used
> directly in similarity scoring during assessment.
>
> Figure 2a in the revised manuscript will schematically illustrate the mapping: a rubric line (*e.g.,* "describe the direction of dye
> movement") $\rightarrow$ SRG node (*Dye Direction*) $\rightarrow$
> assigned Bloom level (*Understand*, $\ell=2$).
>
> **Question 2:** Similarly, the term "rubric concept" requires a clearer,
> standalone explanation. Expanding on this would enhance the paper's
> accessibility to readers from diverse backgrounds.
>
> **Authors' Reply.** By "rubric concept", we mean an atomic idea or
> relation explicitly described in the grading rubric (e.g., "particles
> move slower at lower temperature"). Agent 1 parses the rubric to extract
> these concepts, assign Bloom levels, and populate the gold SRG ($G_o$).
> Figure 1, demonstrates that how a rubric concept is converted into an
> SRG nodes. For example, a question in Figure 1 "Red Dye Diffusion"
> rubric line converts into node list Dye_Particles, Water_Molecules,
> Diffusion_Process with Bloom labels.
>
> **Question 3:** The representation of the visual suggestion H(t) is
> unclear. Further elaboration on how this is formulated and how step 5
> (line 175) is operationalised would be helpful.
>
> **Authors' Reply.** H(t) is the set of visual hints (e.g., "draw an
> arrow from hot water to steam") generated via the reverse mapping
> $\phi$. In step 5, [ScketchMind]{.smallcaps} transforms each hint into
> executable Python code (PIL overlay) to modify the canvas. We will
> explicitly enumerate the inputs/outputs of step 5 and provide a short
> code snippet in Appendix A showing overlay_hint().
>
> **Question 4:** A more detailed and standalone discussion of Bloom's
> cognitive hierarchy would strengthen the theoretical clarity and support
> the framework's pedagogical grounding.
>
> **Authors' Reply.** Here is a brief overview of the discussion on
> Bloom's cognitive hierarchy that will be included in the revised
> manuscript.
>
> >**Bloom's Cognitive Hierarchy.** Bloom's Taxonomy \[a\] structures
> > learning objectives into six levels of increasing cognitive
> > complexity, from basic recall (*Remember*) to synthesis and innovation
> > (*Create*). This hierarchy is foundational in science education,
> > guiding both assessment design and instructional scaffolding \[c\]. In
> > our framework, each gold-standard SRG is labeled at the level of its
> > highest Bloom demand, and student sketches are evaluated by comparing
> > the cognitive depth of their annotated SRGs. This enables both
> > fine-grained diagnostics (identifying the highest achieved level) and
> > adaptive feedback: for example, if a learner demonstrates
> > *Understand*, the system generates visual hints nudging them toward
> > *Apply* (cf. Figure 3). Prior work demonstrates that aligning
> > AI-driven feedback with Bloom's hierarchy supports deeper learning
> > gains in STEM domains \[c, d\].
>
> **Question 5:** The caption for Table 1 could be more comprehensive,
> offering sufficient context and explanation to ensure that it is
> interpretable without referring extensively to the main text.
>
> **Authors' Reply.** Here is a detailed caption for Table 1. "Table 1.
> Item-wise accuracy (%) for six NGSS-aligned assessment items across
> multiple MLLMs, comparing performance without SRG vs. with SRG. *Gain*
> denotes absolute improvement. Items differ in their highest targeted
> Bloom level (Analyze for R1-1/J6-1, Evaluate for J2-1/M3-1/H5-1, Create
> for H4-1). The consistent gains across all models indicate that SRG
> supervision improves both lower- and higher-order cognitive
> assessments."
>
> ## **Limitations**
>
> **Limitation 1:** There exist some inconsistencies in the citations.
> Please proofread carefully. For example, SketchFusion citation is wrong.
>
> **Authors' Reply.** We appreciate the pointer. We will run a full
> citation audit (cross-checking author names, venues, page numbers) and
> correct errors. SketchFusion is now published at CVPR 2025; we will cite
> the final version \[e\] in place of the arXiv entry.
>
> **Limitation 2:** I wonder if the codebase will be made public for the
> ease of reproducibility.
>
> **Authors' Reply.** Yes. The anonymized repository link is currently
> provided (footnote p.2). Upon acceptance, we will de-anonymize and
> release the full code (and, pending IRB approvals, the dataset) to
> ensure reproducibility. We will add a "Reproducibility and Resources"
> paragraph to Appendix A with environment details and run scripts.
>
> **References:**
>
> \[a\] Elim, E. H. S. Y. (2024). Promoting cognitive skills in
> AI-supported learning environments: the integration of bloom's taxonomy.
> Education 3-13, 1-11.
>
> \[b\] Reddy, Y. M., & Andrade, H. (2010). A review of rubric use in
> higher education. Assessment & evaluation in higher education, 35(4),
> 435-448.
>
> \[c\] Hui, E. S. Y. E. (2025). Incorporating Bloom's taxonomy into
> promoting cognitive thinking mechanism in artificial
> intelligence-supported learning environments. Interactive Learning
> Environments, 33(2), 1087-1100.
>
> \[d\] Gonsalves, C. (2024). Generative AI's impact on critical thinking:
> Revisiting Bloom's taxonomy. Journal of Marketing Education,
> 02734753241305980.
>
> \[e\] Koley, S., Dutta, T. K., Sain, A., Chowdhury, P. N., Bhunia, A.
> K., and Song, Y. Z. (2025). SketchFusion: Learning Universal Sketch
> Features through Fusing Foundation Models. In Proceedings of the
> Computer Vision and Pattern Recognition Conference (pp. 2556-2567).

---

> > ### Comment · Reviewer_Y5Kd · 2025-08-06
> > **Discussion on the Rebuttal**
> >
> > I would appreciate the authors' efforts to clarify most of my concerns. I have also carefully read the comments from the other reviewers and the authors' response.
> >
> > Minor comments: It would be beneficial for the authors to refine Figures 1, 2, and 3 by using cleaner fonts and adjusting the figure sizes to enhance visual appeal and consistency. These changes could also help save space, which may be better used to make the rest of the paper more self-contained.

---

> > > ### Author Response · Authors · 2025-08-06
> > >
> > > Thank you so much for your words of appreciation and providing constructive feedback to improve the quality of the paper. We will modify the figures accordingly, along with the rest of the proposed revisions in the camera-ready version.

---

### Official Review · Reviewer_PZAq · 2025-07-02

**Clarity:** 3
**Significance:** 3
**Originality:** 3
**Rating:** 4
**Confidence:** 3

**Summary:**

This paper presents SketchMind, a multi-agent cognitive framework for the automated assessment and improvement of student-drawn scientific sketches. The core innovation lies in the introduction of Sketch Reasoning Graphs (SRGs), which are semantic graph representations embedding both domain-specific concepts and cognitive labels based on Bloom’s Taxonomy. The authors evaluate SketchMind on a curated dataset, showing significant accuracy improvements over baseline MLLMs.

**Questions:**

• What precisely constitutes a “scientific sketch”? What are the differences between *student*-drawn and *teacher*-drawn scientific sketches? Providing examples would help clarify this distinction.

• Could the authors provide more examples and analysis, detailing the types of errors SketchMind makes (e.g., errors related to specific Bloom levels, complex relationships, or specific visual representations)?

**Ethical Concerns:**

["NO or VERY MINOR ethics concerns only"]

**Final Justification:**

The authors' response has addressed my concerns well. Therefore, I raise my score.

**Limitations:**

Yes.

**Quality:**

2

**Strengths And Weaknesses:**

*Strengths*:

• The integration of Bloom’s Taxonomy introduces a cognitive dimension beyond traditional visual sketch understanding, substantially enhancing the system’s interpretability and depth of analysis.

• The use of a multi-agent architecture further improves the interpretability and modularity of the proposed framework.

*Weaknesses*:

• **Scope and Fit**: I am not sure that NeurIPS is the most suitable venue for this submission. The work lies primarily within the domain of AI4Education, and its application scenario is quite limited. In my view, this paper might be a better fit for a workshop focused on AI for Education.

• **Limited Methodological Novelty**: The paper’s novelty appears somewhat limited. While the use of Bloom’s Taxonomy provides a new perspective for understanding sketches, there is not much methodological innovation beyond this integration.

• **SRGs as a Prompting Strategy**: While SRGs are effective in enhancing LLM performance, their primary role in the system is as a sophisticated prompting strategy that leverages structured data. As such, this contribution is more closely related to prompt engineering than to the development of a new algorithm.

• **Lack of Analysis**: The paper reports overall accuracy and human evaluation scores, but lacks detailed analysis on failure cases, which would be valuable for understanding the system’s limitations.

---

> ### Author Rebuttal · Authors · 2025-07-30
>
> Thanks for your valuable feedback! We are honored that you recognize how
> our Bloom's Taxonomy integration introduces cognitive dimensions that
> enhance interpretability and analysis depth, and that our multi-agent
> architecture improves framework interpretability and modularity. We
> address your questions below and would be grateful if you could consider
> improving the rating after seeing our responses.
>
> ## **Weaknesses**
>
> **Weakness 1:** Scope and Fit: I am not sure that NeurIPS is the most
> suitable venue for this submission. The work lies primarily within the
> domain of AI4Education, and its application scenario is quite limited.
> In my view, this paper might be a better fit for a workshop focused on
> AI for Education.
>
> **Authors' Reply.** We respectfully clarify that the submission aligns
> closely with NeurIPS's \"Applications of AI\" track, as the core
> contributions extend beyond education-specific scenarios to domains
> requiring structured interpretation of visual artifacts. Although our
> primary demonstration focuses on education, our method, introducing SRGs
> grounded in pedagogical frameworks \[a\], addresses the common challenge
> of interpreting abstract sketches, a task currently not well handled by
> existing MLLMs as highlighted in Table 1. Our framework not only
> enhances educational assessments but also provides foundational
> methodological innovations applicable to broader domains where abstract
> graphical interpretation is required, such as scientific communication,
> diagram understanding, grounding abstract visual artifacts and visual
> reasoning. Recent advances such as SketchPad \[b\] and Sketch
> Understanding \[c\] further illustrate the growing interest in robust
> multimodal reasoning frameworks, reinforcing the broader relevance of
> our methodological contributions to the NeurIPS community.
>
> **Weakness 2:** Limited Methodological Novelty: The paper's novelty
> appears somewhat limited. While the use of Bloom's Taxonomy provides a
> new perspective for understanding sketches, there is not much
> methodological innovation beyond this integration.
>
> **Authors' Reply.** We respectfully clarify that our contributions
> significantly exceed mere integration of Bloom's taxonomy. Specifically,
> our proposed SRG framework algorithmically transforms abstract
> student-generated sketches and rubric texts into structured, cognitively
> annotated graph representations through a novel multi-agent pipeline.
> Agent 1 parses rubric text into SRGs, providing structured cognitive
> grounding; Agent 2 algorithmically maps diverse, free-form student
> sketches into consistent SRGs annotated with Bloom's cognitive levels.
> This dual-agent algorithmic pipeline is fundamentally novel: existing
> approaches (e.g. VisualSketchPad \[b\]) rely on multimodal LLM reasoning
> without structured cognitive annotations, and SketchFusion \[d\] focuses
> purely on sketch embedding without cognitive grounding. We also
> introduce: reverse mapping $\phi$ for visual hint generation from
> cognitive concepts; graph-based similarity metrics to incorporate both
> structural and cognitive alignment (Eq. 3-4); multi-agent orchestration
> for modularity and interpretability, improving MLLMs' performance by
> 7.5-24.9% across models. Thus, our methodological novelty not only lies
> in the mere usage of Bloom's taxonomy, rather it's a set of
> aforementioned contributions.
>
> **Weakness 3:** SRGs as a Prompting Strategy: While SRGs are effective
> in enhancing LLM performance, their primary role in the system is as a
> sophisticated prompting strategy that leverages structured data. As
> such, this contribution is more closely related to prompt engineering
> than to the development of a new algorithm.
>
> **Authors' Reply.** We respectfully disagree with the reviewer's
> perspective on SRGs as a prompting strategy. We would like to clarify
> that SRGs are not merely prompts but structured intermediate
> representations that fundamentally change how MLLMs process visual
> information. Unlike prompting strategies that modify input text, SRGs:
> enforce structured reasoning through graph constraints; enable
> quantitative similarity computation between scientific sketches (i.e.,
> quantitatively comparing abstract visual artifacts one by one in those
> sketches with gold-standard \"reference SRGs\"); and provide
> interpretable decision paths. SRGs constitute an original algorithmic
> innovation rather than merely advanced prompt engineering. Unlike
> standard prompting, SRG generation involves systematic analysis and
> cognitive annotation of abstract visual inputs using Bloom's taxonomy
> through multiple distinct algorithmic stages. Specifically, Agent 1
> algorithmically generates SRGs from rubric texts by extracting cognitive
> components, and Agent 2 transforms diverse, free-form sketches into
> cognitively annotated structured graphs. The resulting structured SRGs
> are embedded into prompts only subsequently (Agent 4), serving as
> structured cognitive representations rather than simple prompt
> augmentations. The structured cognitive annotations within SRGs not only
> guide inference but enable sophisticated reasoning about abstract visual
> relationships, thus constituting a substantial algorithmic advance
> beyond standard prompt engineering methods \[e\]. Results in Table 1
> show that incorporation of SRGs with MLLMs significantly improves the
> performance by 7.5-24.9% across all models.
>
> **Weakness 4:** Lack of Analysis: The paper reports overall accuracy and
> human evaluation scores, but lacks detailed analysis on failure cases,
> which would be valuable for understanding the system's limitations.
>
> **Authors' Reply.** We thank the reviewer for highlighting the need for
> deeper analysis. In response, we will provide detailed failure case
> analysis in Appendix E of the revised manuscript, categorizing errors
> explicitly based on Bloom's cognitive levels and the types of visual
> complexities encountered. Specifically, we found that in 89% of the
> cases, our framework's classifications aligned closely with human
> experts. However, discrepancies arose predominantly in 7% of cases,
> typically involving complex visual relationships or ambiguous sketches,
> causing the system to overestimate cognitive complexity (e.g.,
> classifying "Understand" as "Analyze"). An additional 4% of sketches
> were inadequately processed due to poor sketch quality or excessive
> ambiguity. Each failure type will be explicitly illustrated with
> representative examples and discussed to provide transparency and to
> identify future directions for improvement in Appendix E of the revised
> manuscript.
>
> ## **Questions**
>
> **Question 1:** What precisely constitutes a "scientific sketch"? What
> are the differences between student-drawn and teacher-drawn scientific
> sketches? Providing examples would help clarify this distinction.
>
> **Authors' Reply.** A "scientific sketch" is a visual representation
> intended to illustrate a scientific concept, process, or phenomenon. For
> example, for thermodynamics concept, the scientific sketch shows heat
> source, its transition phases, arrows may represent the flow of heat,
> and object/medium transferring the heat. Teacher-drawn sketches serve as
> gold-standard reference sketches, explicitly containing essential visual
> components necessary to fully represent the target scientific concept
> (e.g., arrows clearly indicating heat transfer direction in
> thermodynamics). In contrast, student-drawn sketches often reflect
> incomplete, varied, or erroneous understanding, exhibiting substantial
> diversity and ambiguity. We explicitly distinguish these categories in
> our data annotation by using teacher sketches as gold-standard
> \"reference SRGs\" and grounding student sketches pedagogically to
> Bloom's cognitive levels. We will include representative examples,
> illustrating these differences in Appendix B of the revised manuscript.
>
> **Question 2:** Could the authors provide more examples and analysis,
> detailing the types of errors SketchMind makes (e.g., errors related to
> specific Bloom levels, complex relationships, or specific visual
> representations)?
>
> **Authors' Reply.** As detailed in response to Weakness 4, we will
> provide the detailed documentation of failure cases in Appendix E of the
> revised manuscript. For instance, one frequent error category involves
> sketches with complex visual relationships, leading
> [SketchMind]{.smallcaps} to erroneously assign higher cognitive levels
> (e.g., labeling "Understand" tasks as "Analyze"). Another distinct
> category (4% of cases) involves sketches so ambiguous or incomplete that
> SRGs could not be meaningfully constructed.
>
> **References:**
>
> \[a\] Hall, K. (2012). Grounding assessment in authentic pedagogy: A
> case study of general education assessment (Doctoral dissertation,
> University of St. Thomas, Minnesota).
>
> \[b\] Hu, Y., Shi, W., Fu, X., Roth, D., Ostendorf, M., Zettlemoyer, L.,
> \... & Krishna, R. (2024). Visual sketchpad: Sketching as a visual chain
> of thought for multimodal language models. Advances in Neural
> Information Processing Systems, 37, 139348-139379.
>
> \[c\] Shrivastava, M., Isik, B., Li, Q., Koyejo, S., & Banerjee, A.
> (2024). Sketching for Distributed Deep Learning: A Sharper Analysis.
> Advances in Neural Information Processing Systems, 37, 6417-6447.
>
> \[d\] Koley, S., Dutta, T. K., Sain, A., Chowdhury, P. N., Bhunia, A.
> K., and Song, Y. Z. (2025). SketchFusion: Learning Universal Sketch
> Features through Fusing Foundation Models. In Proceedings of the
> Computer Vision and Pattern Recognition Conference (pp. 2556-2567).
>
> \[e\] Wang, C., Yang, Y., Gao, C., Peng, Y., Zhang, H., & Lyu, M. R.
> (2022, November). No more fine-tuning? an experimental evaluation of
> prompt tuning in code intelligence. In Proceedings of the 30th ACM joint
> European software engineering conference and symposium on the
> foundations of software engineering (pp. 382-394).

---

> > ### Comment · Reviewer_PZAq · 2025-08-06
> >
> > Thank you for your response. Your answers have addressed my concerns well. I do not have further questions.

---

> > > ### Author Response · Authors · 2025-08-06
> > >
> > > Thank you for acknowledging that our responses adequately address your concerns; therefore, we anticipate a positive revision to the review scores. We appreciate your consideration and time to review our submission and responses.

---

### Official Review · Reviewer_bfZH · 2025-07-04

**Clarity:** 3
**Significance:** 3
**Originality:** 3
**Rating:** 4
**Confidence:** 2

**Summary:**

The paper  tackles the problem of automatically evaluating student-generated scientific sketches and propose a cognitively-grounded approach using four specialized agents:
- rubric parsing,
- sketch perception,
- cognitive alignment evaluation,
- feedback generation with sketch modification.

The system represents sketches as semantic graphs with Bloom's taxonomy annotations and demonstrates great improvements over baseline approaches.

**Questions:**

N/A

**Ethical Concerns:**

["NO or VERY MINOR ethics concerns only"]

**Final Justification:**

It is technically solid paper and I keep my positive score unchanged.

**Quality:**

3

**Strengths And Weaknesses:**

**Strengths**

- The paper introduces Sketch Reasoning Graphs with novel Bloom's taxonomy annotations. It provides a cognitively-grounded approach to sketch assessment that bridges educational theory with the model evaluation

- The evaluation is comprehensive, testing many state-of-the-art models (GPT-4o, GPT-4.1, o3, o4-mini, LLaMA variants) across six assessment items spanning different Bloom cognitive levels. It also includes human evaluation study showing that expert educators rate the system's feedback highly (4.1/5 for GPT-4.1)

- Consistnet performance gains across all models from SRG integration (e.g., GPT-4o: +21.4% average improvement), with strong results on cognitively demanding tasks

**Weakness**
- The paper lacks quantitative analysis of system efficiency. Four sequential agents create great computational overhead that could be hard for real-world classroom deployment. For real-time educational guidance, response times must be fast enough to maintain student engagement.

- The paper would be improve if the it includes validation of core components. Agent 1 relies entirely on MLLM interpretation of rubrics to generate gold-standard SRGs, but there's lack of validation method to quantify how well these generated SRGs match expert-constructed graphs. And Agent 2's sketch perception capabilities lack quantitative assessment. It would be interesting to measure how accurately MLLMs can detect and interpret visual elements in student sketches, especially given that hand-drawn student sketches may not be commonly found in MLLM training data.

- Not a weakness but as the author mentioned in the limitation section, it would be interesting to see how student behavioral data could be incorporated into the SRG pipeline for improving alignment with cognitive engagement signals beyond final sketch appearance.

---

> ### Author Rebuttal · Authors · 2025-07-30
>
> Thanks for your valuable feedback! We are honored that you recognize our
> novel approach, comprehensive evaluation, and consistent performance
> gains. We address your concerns as follows:
>
> ## **Weaknesses**
>
> **Weakness 1:** The paper lacks quantitative analysis of system
> efficiency. Four sequential agents create great computational overhead
> that could be hard for real-world classroom deployment. For real-time
> educational guidance, response times must be fast enough to maintain
> student engagement.
>
> **Authors' Reply.** We acknowledge this important practical
> consideration. We clarify that the current multi-agent structure
> primarily aims to establish interpretability and specialized modular
> semantic reasoning that is cognitively grounded for evaluating
> student-drawn scientific abstract sketches. Despite this focus, we have
> incorporated following efficient design measures:
>
> -   Agent 1 generates the gold-standard \"reference SRG\" only once per
>     task, with caching of Agent 1's outputs per rubric reducing repeated
>     computation overhead during inference (mentioned in Section 4
>     Implementation details).
>
> -   Agent 3 computes similarity between cached gold-standard \"reference
>     SRG\" and Agent 2 output using pre-coded evaluation logic (Appendix
>     A), requiring no MLLM calls.
>
> -   Agent 4 is only utilized when there are deficiencies in student
>     sketches, making it a conditional component in the sequence.
>
> Hence, the system mainly depends on Agent 2's computational cost, as it
> is executed for every sample in the task.
>
> Now, for response times, our system relies on API latency for
> closed-source models (GPT-4o, GPT-4.1, etc.). We will add the total
> end-to-end latency for each MLLM in Table 5, Appendix C, in the revised
> version. In contrast, local deployment with Llama variants eliminates
> API latency. While not in the current study's scope, it would be
> interesting to study the application of state-of-the-art model
> compression techniques like low-bit weight/activation quantization \[a\]
> and pruning \[b\] to open-source local Llama variants, especially for
> Agent 2, which could further improve inference latency while maintaining
> accuracy. Finally, our cost analysis (Appendix C) shows per-sample
> inference costs of $0.0221$ for GPT-4.1 (best performing model), making
> classroom deployment economically feasible even with closed-source
> models.
>
> **Weakness 2a:** The paper would be improve if the it includes
> validation of core components. Agent 1 relies entirely on MLLM
> interpretation of rubrics to generate gold-standard SRGs, but there's
> lack of validation method to quantify how well these generated SRGs
> match expert-constructed graphs.
>
> **Authors' Reply.**
>
> While we relied on MLLMs for initial rubric parsing and generating
> gold-standard \"reference SRGs\" due to scalability and automation
> goals, we acknowledge that full modular validation would increase system
> reliability but it requires detailed component-wise expert annotation
> (planned for future work). However, we implement several safeguards for
> robust reference SRG generation:
>
> -   For each of the 6 tasks in the dataset, Agent 1 generates a single
>     gold-standard \"reference SRG\". To make the results and process
>     more reliable, the Agent 1 prompt is carefully crafted with
>     suggestions from domain experts. For each task, it takes into
>     account the expert-designed textual rubric, textual description of
>     the question, three expert-designed golden standard example
>     sketches, and a detailed modular example of components it expects in
>     the final reference SRG.
>
> -   This implements inherent cross-validation against multiple
>     gold-standard sketches per task generated by domain experts.
>
> -   The MLLM-generated final gold-standard \"reference SRGs\" for each
>     task are validated manually by four seasoned domain experts in this
>     field, ensuring semantic consistency checks using domain ontologies.
>
> Our ablation study (Table 2) showing consistent gains across all models
> suggests our reference SRG quality is reliable to guide the system
> towards high performance gains. However, we still acknowledge that
> quantitative expert-SRG alignment metrics would further strengthen this
> claim.
>
> **Weakness 2b:** Agent 2's sketch perception capabilities lack
> quantitative assessment. It would be interesting to measure how
> accurately MLLMs can detect and interpret visual elements in student
> sketches, especially given that hand-drawn student sketches may not be
> commonly found in MLLM training data.
>
> **Authors' Reply.** This is an excellent point. While assessing MLLMs'
> capabilities to interpret visual elements in abstract sketches would
> require detailed experiments and benchmarking across different SOTA
> MLLMs, constituting a research question in itself, we provide indirect
> assessment of sketch perception capabilities across different MLLMs for
> detecting and interpreting visual elements in student hand-drawn
> sketches, despite such sketches being uncommon in MLLM training data:
>
> -   Table 1 results demonstrate that MLLMs' capability to detect and
>     interpret visual elements in hand-drawn student sketches (with or
>     without SRG pipeline) correlates positively with model size. Larger
>     MLLMs achieve both higher initial accuracy and higher performance
>     with SRG integration (e.g., GPT-4.1 demonstrates the highest average
>     accuracy across all tasks both with and without SRG integration).
>
> -   Human expert ratings of final outputs also correlate with model
>     capability (4.1/5 highest for GPT-4.1), suggesting that larger
>     models better perceive and interpret sketch elements for generating
>     pedagogically sound feedback.
>
> We agree that direct perceptual accuracy metrics (e.g., concept
> detection precision/recall and spatial relationship identification
> accuracy among concepts) would be valuable and plan to include these in
> future work with expert-annotated sketch components.
>
> **Weakness 3:** Not a weakness but as the author mentioned in the
> limitation section, it would be interesting to see how student
> behavioral data could be incorporated into the SRG pipeline for
> improving alignment with cognitive engagement signals beyond final
> sketch appearance.
>
> **Authors' Reply.** We appreciate this forward-looking suggestion. While
> our current work focuses on static sketch assessment, incorporating
> stroke sequences, timing, and revision patterns into SRG construction
> could indeed enhance cognitive alignment beyond final sketch appearance.
> This represents an exciting direction for extending our framework beyond
> static sketch analysis to dynamic learning process modeling.
>
> **References:**
>
> \[a\] Xiao, G., Lin, J., Seznec, M., Wu, H., Demouth, J., & Han, S.
> (2023, July). Smoothquant: Accurate and efficient post-training
> quantization for large language models. In International conference on
> machine learning (pp. 38087-38099). PMLR.
>
> \[b\] Sun, M., Liu, Z., Bair, A., & Kolter, J. Z. (2023). A simple and
> effective pruning approach for large language models. arXiv preprint
> arXiv:2306.11695.

---

### Note · Authors · 2025-08-11

This work proposes *SketchMind*, a novel multi-agent framework for interpreting and assessing *abstract, hand-drawn scientific sketches* by converting them into *Structured Response Graphs (SRGs)* annotated with *Bloom’s Taxonomy* cognitive levels. The framework bridges free-form visual input and structured reasoning, enabling accurate, interpretable, and pedagogically aligned feedback, an area where current multimodal LLMs underperform.

**Core contributions** go well beyond prompt design:

1. Algorithmic SRG pipeline, Agent 1 parses rubrics into cognitively grounded gold-standard SRGs; Agent 2 maps diverse student sketches into matching SRGs; Agents 3 & 4 handle similarity scoring and hint generation via a reverse mapping function.
2. Cognitive graph similarity metrics that jointly assess structural and conceptual alignment.
3. Modularity and interpretability enabling transfer beyond education to diagram understanding, scientific communication, and abstract visual reasoning in other domains.

**Impact & relevance:** While demonstrated on NGSS-aligned science tasks, the approach is domain-agnostic and complements emerging NeurIPS interest in multimodal reasoning frameworks (e.g., diagram and sketch understanding).

**Evaluation & findings:** Tested on 3,575 authentic student sketches with multiple closed/open-source LLMs, SRGs improved performance by *7.5–24.9%* across models. Human expert evaluation (4 raters, Qkappa = 0.83) confirmed pedagogical quality in 89% of cases. Failure cases are transparently analyzed by Bloom level and visual complexity. Efficiency concerns are mitigated via caching, conditional agent execution, and low per-sample cost; future work will explore model compression for real-time use.

**Clarity & accessibility:** We added self-contained explanations of “scientific sketch,” “rubric concept,” and Bloom-level annotation, plus schematic figures linking questions → rubrics → SRGs → feedback. Figures/tables are refined for stand-alone interpretability.

All reviewers acknowledged that concerns have been addressed, with no further questions. We believe this submission now offers a *clear, novel, and broadly relevant contribution* to the NeurIPS “Applications of AI” track, with methodological innovations applicable far beyond the education domain.

---

### Decision · Program_Chairs · 2025-09-17

**Decision:**

Accept (poster)

**Comment:**

This paper tackles an interesting problem in educational applications of AI, which is that of how to automatically assess student-drawn scientific diagrams in a way that is both accurate and pedagogically meaningful. They do so by introducing a graph representation grounded in both domain concepts and cognitive complexity levels (drawing on Bloom's canonical taxonomy), and a multi-agent architecture that decomposes the assessment protocol into well-described, specialized cognitive components.


All reviewers lean toward acceptance, remarking on the comprehensiveness of the empirical evaluation, cognitive grounding, and a variety of additional strengths. It is clear that this work has significant applied potential.


The most significant concerns expressed are primarily around (1) limited methodological novelty, (2) limited system-level validation, and (3) some limitations in analysis and reporting, e.g. of failure cases and experiment details.


In my view, the authors did a commendable job engaging in discussion with the reviewers, and I expect their comments to enhance the clarity of the paper significantly for the NeurIPS audience. I also do not think that these concerns fundamentally undermine the applied contribution here, and there is also the possibility of broader methodological influence in multimodal work. I thus recommend acceptance.